# Fetal Brain Damage during Maternal COVID-19: Emerging Hypothesis, Mechanism, and Possible Mitigation through Maternal-Targeted Nutritional Supplementation

**DOI:** 10.3390/nu14163303

**Published:** 2022-08-12

**Authors:** Chiara Germano, Alessandro Messina, Elena Tavella, Raffaele Vitale, Vincenzo Avellis, Martina Barboni, Rossella Attini, Alberto Revelli, Paolo Zola, Paolo Manzoni, Bianca Masturzo

**Affiliations:** 1Department of Maternal, Neonatal and Infant Medicine, University Hospital “Degli Infermi”, 13875 Ponderano, Italy; 2Sant’Anna Hospital, Department of Surgical Sciences, University of Turin, 10126 Turin, Italy; 3Sant’Anna Hospital, Department of Public Health and Pediatric Sciences, University of Turin, 10126 Turin, Italy

**Keywords:** COVID-19, pregnancy, fetal brain damage, nutrition, supplementation in pregnancy, prevention

## Abstract

The recent outbreak of the novel Coronavirus (SARS-CoV-2 or CoV-2) pandemic in 2019 and the risk of CoV-2 infection during pregnancy led the scientific community to investigate the potential negative effects of Coronavirus infection on pregnancy outcomes and fetal development. In particular, as CoV-2 neurotropism has been demonstrated in adults, recent studies suggested a possible risk of fetal brain damage and fetal brain development impairment, with consequent psychiatric manifestations in offspring of mothers affected by COronaVIrus Disease (COVID) during pregnancy. Through the understanding of CoV-2’s pathogenesis and the pathways responsible for cell damage, along with the available data about neurotropic virus attitudes, different strategies have been suggested to lower the risk of neurologic disease in newborns. In this regard, the role of nutrition in mitigating fetal damages related to oxidative stress and the inflammatory environment during viral infection has been investigated, and arginine, n3PUFA, vitamins B1 and B9, choline, and flavonoids were found to be promising in and out of pregnancy. The aim of this review is to provide an overview of the current knowledge on the mechanism of fetal brain damage and the impact of nutrition in reducing inflammation related to worse neurological outcomes in the context of CoV-2 infections during pregnancy.

## 1. Introduction

Viral infections during pregnancy have been widely studied due to the negative impact on pregnancy outcomes, including effects on the central nervous system (CNS), intrauterine growth restriction (IUGR), increased infant mortality, congenital disease, miscarriage, and a higher risk of preeclampsia [1].

Due to immunologic alteration and reduction in adaptive immune responses, pregnancy status is related to an impaired clearance of pathogens and increased susceptibility to infections [2,3,4]. Moreover, the hormonal changes have a key role in suppressing cell-mediated immunity at the maternal–fetal edge, leading to a higher risk of severe manifestations [5].

The recent outbreak of the novel Coronavirus (SARS-CoV-2 or CoV-2) pandemic in 2019 and the risk of CoV-2 infection during pregnancy led the scientific community to investigate the potential negative effects of Coronavirus infection on pregnancy outcomes and fetal development. In particular, as CoV-2 neurotropism has been demonstrated in adults [6,7,8,9,10], some authors suggested a possible risk of fetal brain damage and fetal brain development impairment, with consequent psychiatric manifestations in offspring of mothers affected by COronaVIrus Disease (COVID) during pregnancy [11,12,13].

In order to predict the potential impact of CoV-2 on the fetal brain, the behavior of other neurotropic viruses can be considered. Among neurotrophic viruses responsible for fetal brain damage, the most common are Cytomegalovirus, Rubella, Parvovirus B19, Human Herpes Viruses 1 and 2, Varicella Zoster Virus, Zika Virus, Dengue Virus, and Human Immunodeficiency Virus. The entity of fetal brain injury, generally, depends on the development stage and the cell population involved and may result in hydrocephalus, seizures, blindness, hearing loss, and developmental delay [14]. Fetal infection, due to a direct access of the virus from maternal blood flow to the amniotic fluid and, then, to fetal cells, presents different percentages according to host immune system status, pathogen features, gestational age, and virulence; however, vertical transmission is not the only way viruses can damage the fetal central nervous system (CNS).

Through the understanding of CoV-2’s pathogenesis and the pathways responsible for cell damage along with the available data about neurotropic virus attitudes, some authors suggested different strategies to lower the risk of neurologic disease in newborns.

In this regard, the role of nutrition in preventing fetal damage in the case of viral infection has been investigated among first-line strategies [15]. Poor or inadequate nutrition contributes to worsening maternal response to infection, and micronutrient deficiencies have negative effects on fetal growth, intellectual development, perinatal mortality, and susceptibility to infections [16].

The aim of this review is to provide an overview of the current knowledge on the mechanism of fetal brain damage and the impact of nutrition in improving neurological outcomes in the context of CoV-2 infections during pregnancy.

## 2. SARS-CoV-2 Vertical Transmission, Fetal Infection, and Potential Brain Damage

The way SARS-CoV-2 enters cells is widely known and involves as the receptor the angiotensin-2 converting enzyme (ACE2), bolstered by transmembrane protease serine 2 (TMPRSS2), which promotes the fusion of the viral capsid with the host cell [17,18] (Figure 1). Several human tissues express ACE2, including placental syncytiotrophoblast cells, and histological analysis of SARS-CoV-2-infected placenta showed direct and indirect clues of infection, with placental lesions and signs of malperfusion and inflammation [19]. This evidence of tissue alterations suggests a weakening of the placental barrier with higher risk of vertical transmission; however, recent studies demonstrated a limited SARS-CoV-2 expression ex vivo in terms of placentae under acute infection and inefficient SARS-CoV-2 placental replication in vitro in different types of placental tissues [20]. One of the few described cases of vertical transmission was reported in a preterm fetus at 29 weeks of gestation, which underwent emergency C-section for abnormal fetal heart rate, a few days after mild maternal COVID [21]; placental examination showed extensive and multifocal chronic intervillositis, with intense cytoplasmic positivity for SARS-CoV-2, and later, the baby developed extensive cystic peri-ventricular leukomalacia. In this case, although ACE2 is expressed in the CNS and may act as the receptor for CoV-2, the brain damage related most likely to placental dysfunction, which led to perinatal asphyxia and hypoxic–ischemic encephalopathy, while a direct aggression of the virus on the fetal brain was not determined. Moreover, it has been demonstrated that ACE2 expression is lower between 6 and 14 weeks of gestation [22], rising during the end of pregnancy, when organogenesis is already completed [23], and the co-expression of ACE-2 and TMPRSS2 is undetectable in placental cells throughout pregnancy [24], reducing the risk of direct effects on the premature fetal CNS. In addition, in a recent systematic review of the literature, no cases of intrauterine transmission of SARS-CoV-2 from mothers with COVID-19 to their fetuses was proven, confirming a very low rate of vertical transmission [12]. Actually, CoV-2 has been reported in placentas of severely COVID-affected women [25,26], suggesting that in asymptomatic and mild symptomatic patients, the placental barrier maintains its integrity and function. Besides, the high variability in transmission may suggest that a series of risk factors is involved, such as the stage of pregnancy when infection occurs, viral load, placental susceptibility, CoV-2 variants, and maternal immune system status (Figure 2).

The rarity of vertical transmission, whose actual incidence remains controversial, implies that CoV-2 may not be directly responsible for brain damage, but indirectly, through the hyperstimulation of the maternal immune system, may act on brain development, increasing the risk of neurodevelopmental disorders in neonates [27].

In support of this hypothesis, a recent prospective study [28] showed undetectable levels of SARS-CoV-2 IgM in blood chord samples of babies born to infected women. This finding demonstrates no vertical transmission, as IgM does not cross the placental barrier due to its large molecular weight, and the eventual presence of IgM in fetal bloodstream would mean a fetal endogenous production after virus direct contact. The authors investigated the immune response and searched for changes in the cellular immune repertoire, demonstrating that pregnant women infected by CoV-2 had reduced T cell numbers, both CD4+ and CD8+, while their neonates did not present such a decline. However, though not occurring in neonates, the authors demonstrated that T cell response in placental tissue was altered, with enhanced T cell function and higher production of cytokines. A specific activity of placental T cells during pregnancy, involving T helper 1 to T helper 2 switch, includes the maintenance of maternal–fetal tolerance [29]: progesterone, estradiol, and prostaglandin D promote a T helper 2 cytokine profile with a less pro-inflammatory environment, which facilitates CoV-2 infection. The subsequent enhancing of specific pro-inflammatory T cell subsets consequent to CoV-2 infection may compromise this mechanism of tolerance, with possible inflammatory response against the fetus. Moreover, they found that SARS-CoV-2 infection altered the transcriptome of the mother and neonate, and the same findings are shared with placental tissue, suggesting that, although CoV-2 does not trigger fetal haemopoietic immune response in the placenta, the infection impacts the neonatal immune system. As expected, pregnant women infected with SARS-CoV-2 had increased systemic concentration of cytokines, mainly IL-8, IL-10, and IL-15, compared to control mothers, but also, their neonates presented the same higher levels of IL-8. The presence of IL-8 in fetal cord blood may originate from the fetal immune system or, more likely, by maternal production, and it is associated with dysregulated immune and non-immune activities, including the activation of neutrophils, along with a higher risk of developing encephalopathies [30]. Besides neutrophil activity, it has been demonstrated that the humoral innate immune response plays a role in COVID19 pathogenesis [31], with increased activity of monocyte, macrophage, and complement activation.

The role of maternal cytokine-associated inflammatory response during pregnancy, even in the absence of a pathogen, has been previously investigated in the literature as the link between viral infection and fetal brain injury [32]. Cytokines may originate from the maternal peripheral immune system and can reach the fetal bloodstream by transplacental passage or may be by placental or fetal production [33]. Studies demonstrated that also fetal glial cells can produce cytokines after an inflammatory stimulus and further suggested that microglia may act as a promoter of damage to oligodendrocytes and neurons through cytokine production [34]. IL6, TNFα, IL8, and IL1β are the main cytokines implied in fetal brain injury. It is quite interesting that these cytokines are the same involved in the fetal inflammatory response syndrome (FIRS) secondary to intrauterine infections, which leads to preterm premature rupture of membranes and preterm birth: studies demonstrated that FIRS and not prematurity seems to be involved in periventricular leukomalacia (PVL) and cerebral palsy occurring in preterm babies [35]. Moreover, a recent study demonstrated that fetal sex may influence the maternal capacity to counteract an inflammatory response: pregnancy with a male fetus showed higher levels of IL1β and a greater nitrosative damage in the presence of an oxidative environment, with a possible higher risk of fetal brain damage in males [36].

The mechanism behind fetal brain damage seems to imply a direct injury of cytokine to oligodendrocytes and neurons and a secondary injury through activation of microglia and astrocyte with the release of free radicals, cytotoxic cytokines, and excitotoxic metabolites. The production of free radicals is liable to promote the oxidative and nitrosative stress of the surrounding neurons and oligodendrocytes, while excitotoxic metabolites, such as glutamate and quinolinic acid, binding to NMDA and glutamate receptors, mediate injury to oligodendrocytes. TNFα is demonstrated to decrease the numbers of oligodendrocyte progenitors by apoptosis and inhibition of their differentiation. This inflammatory environment may produce neuronal impairment, with axonal loss, death of neurons, cytoskeletal damage, and interruption in neurons’ migration from the ventricular zone to the neocortex, with abnormal cortical development [37] (Figure 3).

Recently, it was proposed that the increased levels of cytokines during SARS-CoV-2 infection may interfere with the hormonal environment and, in particular, may lead to a dysfunction in the negative feedback between the hypothalamic–pituitary–adrenal (HPA) axis and immune system and dysregulation in placental endocrine function [38]. Evidence demonstrated that the H1N1 influenza virus may decrease placental production of progesterone, leading to an abnormal inflammatory response [39]. Besides, the unbalanced HPA axis activity and the increased production of glucocorticoids evoked by stressors may act on fetal programming with higher susceptibility to neuroendocrine and psychiatric disorders in adulthood [40]. 

Indeed, apart from the organic brain damages, it has long been known that intrauterine or early-life infections are associated with a wide spectrum of cognitive deficits and neuropsychiatric disorders [41]. Depression and bipolar disorder have already been associated with intrauterine infections, while schizophrenia and autism appear to be influenced by genetic and environmental factors, including maternal infections during pregnancy [42]. It has been suggested that the inflammatory milieu may act with different mechanisms, such as functional reprogramming of innate immune cells in the fetal brain, epigenetic changes in brain development genes, and permanent impairments of synaptic pruning. In a recent review, Figueiredo et al. [11] warned about a potential role of SARS-CoV-2 in triggering autism and schizophrenia in affected mothers’ offspring. The anatomopathological aspect of autistic patients is an overabundance of cortical neurons and connections [43]; the finding of high cytokine levels (IL1β, IL6, and IL17), gliosis in periventricular and subcortical areas, and altered microglial cells responsible for dysregulated synaptic pruning, in fetuses affected by CoV-2, might suggest a higher risk of developing autism in this population. Regarding schizophrenia, intense microgliosis is a distinctive feature of this disorder, along with increased synaptic pruning in the pre-frontal cortex and consecutive reduction in the number of synaptic structures. As previously shown, SARS-CoV-2 infection is related to increased gliosis and synapse pruning mediators, potentially favoring schizophrenia development. Not only infection, but also the timing of maternal infection during pregnancy have a pivotal role in the neurodevelopmental outcome of the newborn. Viral infections during the first trimester have been associated with autistic spectrum disorders [38]. Infections occurring during the second trimester may result in suppressed exploratory behaviors, while late gestational infections seemed to be responsible for preservative behaviors, such as schizophrenia and autistic spectrum disorders, and addictive behaviors, such as obsessive compulsive disorders [44].

## 3. Strategies of Prevention

Among strategies against CoV-2 infection, vaccination plays a crucial role in preventing transmission, infection, and severe disease, and its efficacy and safety have been confirmed also during pregnancy by recent meta-analyses [45]. Individual preventative measures, such as self-isolation, social distancing, and sanitation, are fundamental in association with vaccine in preventing transmission and are included in all public health strategies guidelines [46].

A healthy lifestyle and moderate physical activity have proven effective in preventing weight gain and pregnancy diseases such as gestational diabetes [47]. Actually, gestational diabetes is related to increased susceptibility to viral infection, severe disease, and associated complications [48], while obesity is a negative prognostic risk factor in patients affected by COVID, in particular in the case of concomitant malnutrition or trace element deficiency [49].

## 4. Nutrition and Supplementation in Preventing a Negative Fetal Outcome of SARS-CoV-2 Infection

The last two years have seen an extensive increase in scientific studies exploring the role of feeding habits, nutrition, and nutritional supplementation in preventing severe infection or mitigating the negative effects of COVID, both in mothers and offspring. A healthy maternal diet should include vegetables, fruit, legumes, olive oil, nuts, fish, essential and poly-unsaturated fatty acids, and fiber-rich carbohydrates [50]. In addition, minerals and vitamins are highly sensitive to deficiencies during pregnancy and are mostly supplemented to support physiological gestation and immune system activity [50].

In this context of high energy consumption, viral infection represents a stressing process, resulting in abnormal energy consumption and intensive immune system response, which requires supplementary energy intake; moreover, nutrition is one of the most important factors that impact the immune system [51].

The role of supplementation, along with a balanced diet, may enhance immunity and support the body’s natural defenses such as epithelial barriers, antibody production, and normal inflammatory response [52].

Many micronutrients and macronutrients have been extensively studied to understand their possible role in immune-modulation and prevention of viral infection, as well as protective actions against the abnormal immune response, involved in fetal damage.

Hereafter, we report a comprehensive overview of the recent evidence regarding macronutrients and micronutrients and their specific role in the prevention of viral transmission, infection, and negative fetal outcomes and in immune response modulation.

### 4.1. Amino Acids

Amino acids are involved in different immune processes, such as proliferation and activation of lymphocytes T and B, natural killer cells, and macrophages, regulation of intracellular redox status and gene expression, as well as being a constituent part of antibodies, cytokines, and cytotoxic substances [53]. In particular, arginine has been demonstrated to be involved in CoV-2 replication inhibition through the action of three enzymes (neuronal nitric oxide synthase (NOS), endothelial NOS, and inducible NOS), which catalyze the oxidation of l-arginine to nitric oxide (NO) and L-citrulline [54]. NO is a mediator of host defense, and it is produced in a high quantity by inflammatory cell types upon cytokine stimulation. Moreover, it plays an important role as a mediator in non-adrenergic, non-cholinergic neurotransmission, learning and memory, synaptic plasticity, and neuroprotection [55]. Soy protein, peanuts, walnuts, and fish are relatively rich in L-arginine, with approximately 7% of all amino acids contained in fish and 15% in walnuts [56]. Supplementation of arginine is demonstrated to increase lymphocyte counts [57], and although administration is not recommended in the case of severe sepsis, due to its effects on cardiovascular stability, a dose of 10 g is considered optimal and safe when given before a severe infection [58].

### 4.2. Omega-3-Poly-Unsaturated Fatty Acids

Omega-3, especially eicosapentaenoic acid (EPA) and docosahexaenoic acid (DHA), has a known pivotal role in the modulation of inflammation by improving B cell activity and decreasing cytokines (TNF-alfa, IL-1beta, IL-6, and IL-8) and inflammatory eicosanoids [59]. DHA is involved in a cell-protection mechanism against oxidative insults by inducing inflammatory tolerance [60], while EPA directly contributes to decreasing prostaglandin E2 (PGE2) production [61]. Omega-3 can also improve the CD4/CD8 ratio [62] and reduce COX-2 production, while n-3 poly-unsaturated fatty acid (PUFA_-derived lipid mediators, including protectin D1, are involved in the decrease of viral replication [63]. It has been demonstrated that, due to its anti-inflammatory properties, n-3 PUFA could reduce neurological disorders in newborns, such as autism, related to viral infection during pregnancy [64].

### 4.3. Fatty Acids (S/MCFA)

Short- and medium-chain fatty acids (S/MCFAs) have been demonstrated to act as signaling molecules that regulate the body’s energy balance in pregnancy; moreover, S/MCFAs were found to reduce early pregnancy loss through enhancing maternal phospholipid metabolism and ovarian progesterone synthesis [65]. Emerging evidence on Monolaurin identified this monoglyceride of lauric acid, a medium-chain fatty acid present in coconut oil, as a potential natural antiviral, by dissolving the lipids and phospholipids in the pathogen’s envelope and by interfering with signal transduction in cell replication [66]. In addition, a recent study demonstrated anti-inflammatory action with the decrease of IL-6 and IL-8 levels, and it could promote the restoration of homeostasis by regulating the interferon pathway related to its antiviral activity [67].

### 4.4. Vitamin A (All-Trans-Retinol)

Vitamin A is considered an anti-inflammatory factor involved in mucosal integrity, and although there is no actual evidence, it has been proposed as a treatment for CoV-2 infection, as it can inhibit viral replication by upregulating elements of the innate immune response, making them refractory to productive infection during subsequent rounds of viral replication [68]. Moreover, vitamin A is a cofactor during macrophagic activity (phagocytosis, oxidative processes, and regulation of pro-inflammatory TNF-alpha and IL-2), in the regulation of NK cells, and in the development and differentiation of Th1 and Th2 cells [69]. Due to the possible teratogenic effects associated with high doses of vitamin A, an indication for supplementation is reserved only for the prevention of night blindness in populations with a severe deficiency. The main adverse effects associated with excessive vitamin A intake, particularly at the beginning of the first quarter of pregnancy, are congenital malformations involving the central nervous and cardiovascular systems and spontaneous abortion [70].

### 4.5. Vitamin D (25(OH)D)

Vitamin D deficiency is common among the population with a pandemic proportion, involving 33% of the pregnant women in United States [71] and the 60% of all adults worldwide [72]. Obese women have a higher prevalence of deficiency (35%), regardless of latitude and age, as vitamin D is stored in the adipose tissue [72]. Vitamin D has antimicrobial activity by enhancing innate cellular immunity and producing antimicrobial proteins and antioxidative activity, by reducing the expression of pro-inflammatory cytokines (TNFα and INFγ) and increasing the expression of anti-inflammatory cytokines [73]. Moreover, it promotes the differentiation of monocytes and macrophages, increases their killing activity, and supports antigen presentation. Higher COVID incidence and, more frequently, severe disease were found in patients with vitamin D deficiency [74]. Thus, in the case of vitamin D deficiency during pregnancy, a maximum dose of 5000 IU/day was concluded as safe and effective to achieve the optimal concentration of 25(OH)D [75].

### 4.6. Vitamin E (Tocopherol)

Vitamin E, including α-, β-, γ-, and δ-tocopherol, has a key role as an antioxidant, by limiting the harmful effects of peroxyl radicals on the cellular surface and reducing pathogens’ virulence [76]. α-tocopherol sources are olive and sunflower oils, while γ-tocopherol is found in soybean and corn oil. Regarding immune-modulation, vitamin E is involved in increasing lymphocytes’ proliferation, immunoglobulin levels and response, NK cells’ activity, modulation of Th1/Th2 balance, and IL-2 production. Moreover, it reduces PGE2 production through decreasing the NO levels and inhibiting COX-2, contributing to reducing the pro-inflammatory environment [49]. Concerning effects of vitamin E on viruses, it has been demonstrated that tocopherol reduces the duration of influenza virus infection, suggesting a possible activity against CoV-2 [77]. As mentioned for vitamin A, vitamin E supplementation during pregnancy should be given under strict control, as overdosing can be potentially dangerous.

### 4.7. Vitamin B1 (Thiamine)

Vitamin B1 is an essential micronutrient involved in T cells’ regulation, and its deficiency is associated with inflammation, overexpression of pro-inflammatory cytokines (Il-1, TNFα, and IL-6), and increased levels of eicosanoids derived from arachidonic acid [78]. Pregnant women physiologically require more thiamine, especially during the third trimester; in the case of hyperemesis gravidarum, there is an increased risk for thiamine deficiency due to vomiting. Moreover, it has been demonstrated that thiamine has a critical role in fetal neurodevelopment, as it is involved in the synthesis of myelin and neurotransmitters (e.g., acetylcholine, γ-aminobutyric acid, glutamate), and its deficiency increases oxidative stress by decreasing the production of reducing agents. A lack of thiamine also leads to neural membrane dysfunction, because it is a structural component of mitochondrial and synaptosomal membranes [79]. A recent study, examining the effects of thiamine maternal deprivation in rats, demonstrated a significant decrease in the newborn brain weight, as well as evidence of crucial brain enzyme activity alterations [80]. As thiamine is required for metabolism including that of glucose, amino acids, and lipids and a lack of this vitamin may lead to beriberi and Wernicke encephalopathy in the mother and neurological impairment in the offspring, supplementation during pregnancy is mandatory, especially in the case of hyperemesis, infections, and during the third trimester. Food sources of thiamine include whole grains, legumes, and some meats and fish.

### 4.8. Vitamin B6 (Pyridoxine)

As other group B vitamins, pyridoxine contributes to T cells’ and interleukins’ production and lymphocytes’ maturation, and its deficiency is associated with a decrease in lymphocytes’ proliferation [69].

### 4.9. Vitamin B9 (Folic Acid)

Folic acid is significantly involved in blood cells’ production and immunomodulating properties affecting T cell differentiation [81]. Folate occurs naturally in several foods, including beef liver, leafy vegetables, peas and beans, avocados, eggs, and milk. Default supplementation is strongly recommended during the first trimester and, even, periconceptionally. Folic acid deficiency is associated with fetal malformations, such as neural tube defects and congenital heart disease, and maternal anemia, leukopenia, and thrombocytopenia [15]. Folate is a cofactor for the synthesis of purines and thymidylate and plays a crucial role in DNA synthesis and replication. Moreover, it also contributes to epigenetic processes that influence the phenotype of an organism [82]. A recent meta-analysis evidenced that folate supplementation had a positive impact on offspring’s neurodevelopmental outcomes, including improved intellectual development and reduced risk of autism traits, attention deficit hyperactivity disorder (ADHD), and behavioral and language problems [83]. However, Folate over-supplementation was not associated, in this study, with an improvement in the offspring’s brain development and may have a negative impact on the offspring’s neurodevelopmental outcome [84]. According to international guidelines, the routine universal prophylaxis dose suggested is 0.4 mg per day for 12 weeks in low-risk pregnancy and 4 mg per day in high-risk pregnancy, with a previous neural tube defect [85,86]. Given the possible negative effects following folic acid overdose, supplementation should be considered in the case of viral infection, but with a controlled amount of folic acid, not exceeding 4–5 mg per day [87].

### 4.10. Vitamin B12 (Cobalamin)

Cobalamin is essential for DNA synthesis and regulation and plays a central role in hemopoiesis. Moreover, it can suppress systemic inflammation by modulating IL-6, growth factors, and other anti-inflammatory mediators and is responsible for downregulating nuclear transcription factor-kB (NF-kB) involved in immune response to infection. Vitamin B12 contributes to improving the immune response by increasing CD8 T cells and natural killer T cells against viral infections [69,88]. In addition, it displays antioxidant properties through the reduced glutathione-sparing effect [89]. The recommended daily allowance of vitamin B12 is 2.6 µg/day for pregnant women [88]. Cobalamin deficiency causes megaloblastic or pernicious anemia along with neurologic symptoms occurring in the late stage of disease. A study published in 2020 found that low levels of B12 in pregnant women may increase susceptibility to CoV-2 infection [90]. Besides, in a recent review investigating the role of B12 in CoV-2 infection, cobalamin was found useful as an additional treatment of mild to severe COVID-19 symptoms, because of its analgesic function and role in neuromuscular disorders following CoV-2 infection [88].

### 4.11. Vitamin C (Ascorbic Acid)

Ascorbic acid is a hydrosoluble vitamin, and thanks to the molecular structure, its main role is to protect cells from oxidative injury by donating electrons and to act as a co-factor of eight human enzymes involved in the biosynthesis of collagen, carnitine, hormones, and neurotransmitters. Among immunologic functions, vitamin C improves neutrophil migration and chemotaxis, modulates the increase of phagocytic cells and the maturation of B and T lymphocytes [91], and supports microbial killing and antibody production. In addition, ascorbic acid has been suggested to prevent the excessive activation and accumulation of neutrophils and to decrease epithelial damage [73]. The main dietary sources of vitamin C are vegetables and fruits, particularly citrus fruits, strawberries, potatoes, and tomatoes. Vitamin C deficiency causes scurvy, a polysymptomatic disease consequent to altered collagen synthesis, which impairs tissue reparation and, subsequently, compromises the integrity of tissue barriers. Currently, scurvy is rare in high-income countries, although a less marked deficiency may be present in patients suffering from alcoholism, mental disorders, or unusual eating habits [92], but also in patients with severe lung infections [93]. During pregnancy, vitamin C plasma levels decrease, due to hemodilution and active transfer to the fetus [94], leading to possible mild deficiency. Recently, after the COVID-19 pandemic’s outbreak, several meta-analysis were produced to assess the effectiveness of vitamin C supplementation in the case of severe COVID: the research concluded with no high-quality evidence of reducing the severity and mortality of the disease [95,96,97,98]. However, a recent expert panel document from the National Institute of Health suggests, in the case of supplementation request, a safe dose of 1.5 g/kg body weight [99].

### 4.12. Iron

Iron supplementation during gestation is quite a common practice, due to its fundamental role in hemopoiesis and the high frequency of pregnancy-related anemia. Moreover, iron drives the process of the differentiation and growth of epithelial tissue, and it is involved in the production of reactive oxygen species (ROS) by neutrophils [69]. Although ROS production can enhance the action of the immune system against pathogens, it can also exacerbate infections and inflammation due to oxidative stress. Iron-containing enzymes are essential for viruses, including Coronaviruses, to complete their replication process [100]. Inflammation during SARS-CoV2 infection promotes hepcidin production, which inhibits iron uptake in the gut by binding ferroportin, with a consequent iron sequestration in macrophages, enterocytes, and hepatocytes; SARS-CoV2 may colonize these cells, and the sequestered iron could be employed for viral replication [101]. Emerging studies [102] indicate that iron chelation is a promising adjuvant therapy in treating viral infection and it is plausible that deprivation of iron supply to the virus can be useful also in the case of SARS-CoV-2 infection [100]. Therefore, iron supplementation in pregnancy affected by Sars-CoV-2 can be merely considered in the case of documented iron deficiency.

### 4.13. Zinc

Zinc is a relevant micronutrient involved in preserving the integrity of body homeostatic mechanisms, including immune response, and it is requested for the biological activity of many enzymes and proteins and for cellular proliferation and genomic stability [103]. Animal products such as meat, fish, shellfish, fowl, eggs, and dairy contain zinc, while the food plants containing the most zinc are wheat (germ and bran) and various seeds, including sesame, poppy, alfalfa, celery, and mustard. Zinc is also found in beans, nuts, almonds, whole grains, pumpkin seeds, sunflower seeds, and blackcurrant. Regarding immune system function, zinc is crucial for the development, differentiation, and activation of T lymphocytes [69] and for the secretion of immune-mediating factors, such as IL-1β, IL-6, and TNFα [73]. Moreover, it is implied in NK cells activity, in particular with regard to major histocompatibility complex (MHC) class I [49]; thereby, zinc deficiency may lead to thymic atrophy, T cell lymphopenia, and a reduction of immature B cells with a decrease in antibody production [104]. Zinc contributes also to phagocytosis through NADPH oxidase activity regulation, and it is bound as a divalent cation to metallothioneins (MTs) and released in order to reduce ROS generated by pathogens [49]. In addition, a high intracellular zinc concentration has been demonstrated to inhibit the activity of RNA-dependent RNA polymerase, which results in RNA virus replication impairment. Velthuis et al. demonstrated that zinc inhibits the replication of SARS-Coronavirus (SARS-CoV) in cell culture, by inhibiting the RNA-synthesizing activity of the multiprotein replication and transcription complex and by blocking RNA-dependent RNA polymerase elongation and template binding [105]. In a recent study conducted on pregnant women affected by SARS-CoV2, Anuk et al. demonstrated that zinc levels decreased in COVID patients in all trimesters, compared to healthy pregnant women [106]. The decrease in zinc levels during infections is related to active intracellular transport mediated by zinc transporter ZIP14, induced by IL-6, with an increase in intracellular zinc levels during the acute phase. COVID patients with low zinc levels showed higher complication rates, with prolonged hospital stays and increased mortality, with an odds ratio of 5.54 [107]. Based on this consideration, studies suggested zinc supplementation in the case of viral infections, finding a prophylactic effect and a reduced length of symptoms [108,109,110].

### 4.14. Selenium

Selenium is an essential trace element, and dietary selenium largely comes from meat, nuts (Brazil nuts), cereals, and mushrooms. The U.S. Recommended Dietary Allowance (RDA) of selenium for teenagers and adults is 55 µg/day, and exceeding the Tolerable Upper Intake Level of 400 micrograms per day can lead to selenosis, a poisoned condition characterized by gastrointestinal disorders, hair loss, sloughing of nails, fatigue, irritability, and neurological damage [111].

Small amounts of selenium are essential to produce 25 selenoproteins, with a selenocysteine nucleus, acting as antioxidants, such as glutathione peroxidase (GPx), involved in the control of reactive oxygen species (ROS), deiodinases (Iodothyronine deiodinases), metal chelators (SEPP1), reductases (Thioredoxin reductases, TrxR), modulators of metabolism and insulin sensitivity (SEPS1), and glycoprotein folders (SEP15) [112].

Selenium deficiency is associated with higher levels of inflammation cytokines in various tissues, including the gastrointestinal tract, uterus, and central nervous system. Moreover, deficiency is implied in a lower proliferation of T cells and a decrease in lymphocyte toxicity and NK cells activity, with limited production in ROS by neutrophils, due to insufficient synthesis of glutathione peroxidases, which may facilitate viral replication and virulence by increasing the rate of genome mutation of RNA viruses [113]. In a recent study, Erol et al. showed that pregnant women with COVID in the second and third trimesters had lower selenium levels compared to healthy pregnant patients, and the decrease appeared to be related to IL-6 levels [114]. Zhang et al. demonstrated that low selenium levels were associated with a higher death rate in COVID-19 patients [115], suggesting that selenium could be a choice treatment in SARS-CoV2 disease. On the other hand, besides the possible toxicity, the increase in selenium levels is demonstrated to increase the expression of stress-related selenoproteins and genes involved in inflammation and interferon γ responses [116], sustaining a proinflammatory response.

### 4.15. Copper

Copper is an essential trace element, and dietary copper intake is provided by oysters, beef and lamb liver, Brazil nuts, blackstrap molasses, cocoa, black pepper, lobster, nuts and sunflower seeds, green olives, avocados, and wheat bran. It is absorbed in the gut, bound to albumin, transported to the liver, and finally, distributed to tissues through ceruloplasmin. The RDA for copper during pregnancy is 1000 μg/day, not exceeding 10 mg/day. It is important to note that zinc competes with copper for absorption in the jejunum, and high zinc doses (>150 mg/day) may result in copper deficiency in health individuals [117]. Copper deficiency can produce anemia-like symptoms, neutropenia, bone abnormalities, hypopigmentation, impaired growth, increased incidence of infections, osteoporosis, hyperthyroidism, and abnormalities in glucose and cholesterol metabolism.

Copper is crucial for the immune system response, and it plays an important role in the functions of T helper cells, B cells, neutrophils, natural killer cells, and macrophages [117]. Due to the easy interconversion between the oxidated and reduced form, copper is involved in electrons and oxygen transportation and participates in the production of ROS by macrophages. Cu^2+^ ions are involved in the inhibition of viral entry and replication and the degradation of mRNA and capsid proteins. Cu-ions can inactivate some enveloped or non-enveloped, single- or double-stranded DNA or RNA viruses, and it has been demonstrated that copper can destroy the viral genome and irreversibly affect the morphology of human Coronavirus, including disintegration of the envelope and dispersal of surface Spikes [118].

### 4.16. Magnesium

Magnesium is an enzymatic activator with a central role in various physiological functions such as cell cycle, metabolic regulation, muscle contraction, and vasomotor tone, and its supplementation can reduce lung symptoms, protect the nervous system, improve cardiovascular function, ameliorate liver and kidney injury, and control the blood glucose level by the inhibition of inflammation, oxidative stress, and smooth muscle contraction [119]. Intracellular free magnesium regulates the cytotoxic functions of NK cells and CD8+ T cells, and a decrease in intracellular free magnesium causes defective expression of the natural killer activating receptor NKG2D on NK and CD8+ T cells along with impaired cytolytic responses. In addition, magnesium deficiency enhances oxidative stress and intracellular glutathione depletion, increases the inflammatory cytokine release from monocytes, macrophages, and leukocytes, and increases the susceptibility of tissues to oxidative stress [120]. During pregnancy, magnesium supplementation is given in the case of uterine contractions, preterm delivery, or preeclampsia, in order to provide adequate neuroprotection of the immature fetal brain. A recent study showed a higher magnesium serum level during the first and third trimesters in pregnant COVID patients compared to healthy controls, with a possible negative effect on white blood cell concentrations [106]. A study, carried out in patients with parasite infections, showed that high magnesium levels may reduce nitric oxide production and avoid the microbicidal activity of macrophages, by enhancing Mg^2+^-dependent ecto-APT-ase activity [121]. Moreover, hypermagnesemia seemed to be linked with higher mortality rates in the case of infections and pneumonia [122], and since respiratory muscle weakness is a potential side effect of magnesium sulfate, it should be used judiciously in women with established respiratory distress [123]. On this basis, magnesium supplementation in the case of pregnancy complicated by SARS-CoV-2 infection can be adopted, although hypermagnesemia should be avoided.

### 4.17. Choline

Choline is an essential nutrient obtained through endogenous synthesis, despite the main intake occurring from the diet as a free molecule and in the form of phospholipids, especially as phosphatidylcholines are contained in organ meats, egg yolks, grains, vegetables, fruit, and dairy products [124]. For adult pregnant women, the adequate intake for choline is set at 550 mg/day [125]. Choline has important and different functions in cellular maintenance and growth through the life stages, including neurotransmission (as the precursor for the synthesis of acetylcholine), membrane synthesis (as a basis in the synthesis of phosphatidylcholine), lipid transport, epigenetic regulation of gene expression (through S-adenosyl methionine synthesis and DNA methylation), and one-carbon metabolism (as a methyl group donor, participating in the re-methylation of homocysteine to methionine) [124]. Due to its crucial function in the central nervous system, Freedman et al. demonstrated that higher maternal choline levels during maternal infection are protective for infant brain development and mitigate the adverse effects of inflammation on the offspring’s behavior at 3 months [126]. The same findings have been shown in the case of maternal respiratory virus disease occurring in early pregnancy (<16 weeks of gestational age), when hippocampal inhibitory interneurons are most vulnerable to maternal infections, and neuronal dysregulation in this gestational window may impair infant attention and other self-regulatory behavior [127]. Poor attention in early childhood is considered an early sign of psychopathology and predicts severe mental illness in adults [128]. Maternal choline supplementation in pregnancy has been demonstrated to decrease fetal brain IL-6, reducing inflammation from maternal immune activation, and reversing behavioral effects [129]. Three different double-blinded placebo-controlled studies proved that choline supplementation during pregnancy improves infant cognition, behavior, and social attention, without serious adverse effects [130,131,132]. The role of choline in infant development in SARS-CoV2 infection is yet unknown; however, after the Centers for Disease Control and Prevention [133] warnings on COVID-19 effects on fetal brain development, Hoffmann et al. analyzed choline levels in pregnant women with COVID with attested inflammatory response, defined as an increase in C-reactive protein (CRP) concentrations >10 mg/L. The authors found that infants of infected mothers with choline levels higher than 7.5 μM showed a better score in neonatal reactivity and self-regulation at 3 months of age compared to infants of infected mothers with lower choline levels during the third trimester [134]. On these bases, choline supplementation should be considered in the case of novel Coronavirus disease in pregnancy, in order to mitigate the fetal brain effects of inflammation, and according to the latest reviews, daily intake can be safely raised to 1000 mg/day [135].

### 4.18. Phytonutrients

The term “phytonutrients” defines a compound of substances derived from plants with health benefits. It includes polyphenols, defined as organic complexes with a polyphenolic structure, composed by several hydroxyl groups on aromatic rings. Polyphenols are natural antioxidants mainly contained in green tea, broccoli, and apples and are involved in reducing inflammation and immune response.

Curcumin is a phenolic pigment derived from *Curcuma longa* species plants, with antioxidant, anti-inflammatory, and antifibrotic properties. Antiviral activity has been demonstrated for SARS-CoV2 and other viruses and involves a good binding energy affinity to SARS-CoV-2 main protease (Mpro)/chymotrypsin-like (3CL), with inhibition of viral replication [136]. Moreover, it is implied in decreasing the macrophage population [137] and inhibiting NLR family pyrin domain containing 3 (NLRP3) inflammasome signaling, with a reduction in NFkB, TNFα, IL6, IL1β, and IL18 expression [138]. A recent study, conducted on mice, demonstrated that phytosomal curcumin can attenuate the inflammatory pathology and potentially reverse the detrimental effects of chronic glial activation [139]. Actually, no studies on curcumin supplementation in pregnancy are available in the literature, but the lack of toxicity, even at high doses, ensures a wide-ranging safety profile; nevertheless, the low systemic bioavailability and the rapid metabolism may limit the clinical effects [140].

Flavonoids are a class of polyphenolic secondary metabolites, with antioxidant properties, contained in red wine, oranges, red fruits, and vegetables. They play a role as free-radical scavengers and antioxidants, with anti-inflammatory and anti-viral effects. It has been demonstrated that flavonoids can inhibit influenza virus and Toll-like receptor signaling, by blocking NF-KB translocation [141]. Epigallo-catechin 3 gallate (EGCG) is a catechin belonging to the flavonoids family, and it is the most potent ingredient in green tea with antibacterial, antiviral, antioxidative, anticancer, and chemo-preventive activities [63]. Besides, Kaempferol, contained in some vegetables, such as spinach, cabbage, kale, and broccoli, has demonstrated antioxidant and anti-inflammatory properties, with anti-viral activity due to its high affinity binding with the ACE2 receptor. According to these considerations, the increase in foods containing kaempferol intake during pregnancy may offer health benefits [140]. Quercetin, contained in onion, grapes, shallots, tea, Ginko bilboa, and tomatoes, provides anti-inflammatory activity by inhibiting lipid peroxidation and lipopolysaccharide-induced IL8 production [142], together with an immunosuppressive effect on dendritic cells by downregulating T-helper-induced IL4 [143]. Regarding safety data in pregnancy, no adverse events were reported in clinical trials, and in dietary supplements, recommended daily doses of quercetin usually reach 500–1000 mg [144]. Apigenin, stored in parsley, celery, onions, and oranges, is a flavonoid with antiviral and anti-inflammatory activity, which acts by modulating dendritic cells, responsible for immune balance, decreasing IL6, reducing COX-2 activity, and potentially inhibiting SARS-CoV2 viral replication by Mpro binding [145]. A recent study investigated the therapeutic effect of apigenin on neuroinflammation, demonstrating that chronic apigenin supplementation significantly reduced microglia activation in mice.

According to these findings, supplementation of selected flavonoids, mainly by consuming vegetables rich in these polyphenols, may mitigate negative effects on the fetus in pregnancy affected by COVID.

Table 1 summarizes the evidence for each nutrient considered.

## 5. Conclusions

This revision of the literature provides a comprehensive overview of the current evidence on a wide range of micronutrients, with a focus on the prevention of fetal brain damage.

Following the Centers for Disease Control and Prevention (CDC) warning on fetal brain damage in the case of maternal SARS-CoV2 infection [133], we believed that this emerging topic deserved to be investigated and discussed on the basis of the latest literature. When we started to write this article, we were aware that very little evidence was available on fetal brain damage in the case of maternal COVID: due to the very young age of babies born from mothers affected by COVID and the very little time that has passed since the disease outbreak, only a few scientific studies have been produced, based on hypotheses derived from previous research on different viruses.

For now, we know that SARS-CoV-2 is a neurotropic virus with a specific way of entering cells and is responsible for a massive inflammatory response mediated by cytokines and cells of the innate and adaptative immunity in the host. 

On the other hand, the properties of nutrients have been extensively studied for a long time, and specific antimicrobial and antiviral activities were observed in some of them.

As inflammatory activity was found to be prominent in the case of maternal COVID with a possible indirect damage to fetal cells, including central nervous system cells, supplementation with nutrients, endowed with anti-inflammatory and neuroprotective activity, may be helpful in counteracting the negative effects of CoV-2 infection in pregnancy, with a slight impact on fetal brain development. 

As previously shown, fetal brain damage during SARS-CoV-2 infection looks to be mediated by the maternal immune response and consequent release of ROS, pro-oxidant products, and cytokines, which interfere with normal neurological development and neurons’ migration. A diet rich in antioxidant and anti-inflammatory micronutrients, in our opinion, may have a mild activity in mitigating the negative effects of inflammation on the fetal brain. In particular, arginine, n-3 PUFA, vitamins B1 and B9, choline, and flavonoids have demonstrated an in vivo and in vitro beneficial effect in reducing cellular damage in and out of pregnancy and may be considered in pregnant patients with COVID. 

As side effects have not been studied in pregnant women, a judicious administration following the old-fashioned, Latin principle “in medio stat virtus” may be the right choice until further research sheds light on the efficacy, timing, and doses of the correct supplementation in pregnancy affected by COVID.

## Figures and Tables

**Figure 1 nutrients-14-03303-f001:**
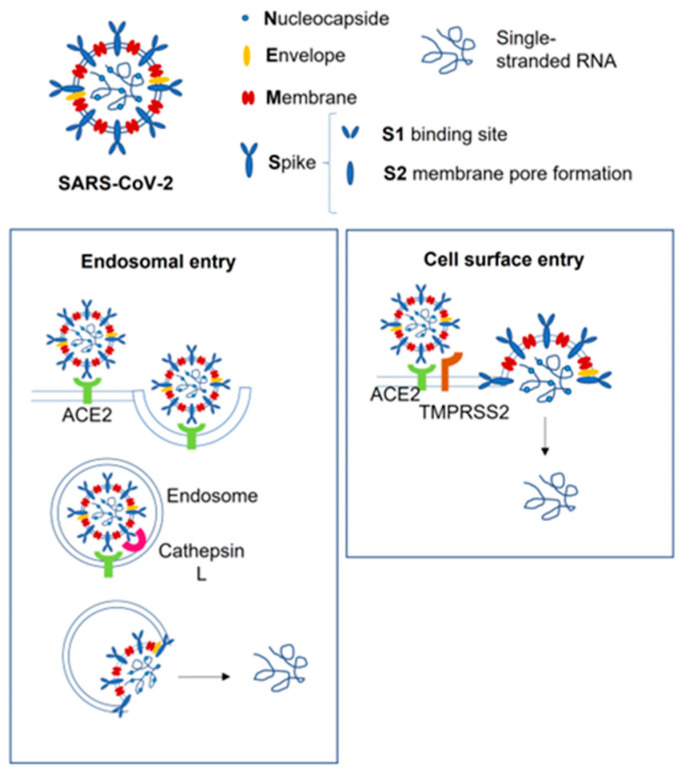
SARS-CoV-2 entry pathways. The cell surface entry pathway requires Spike viral protein to bind ACE2 bolstered by TMPRSS2, which promotes the fusion of the viral capsid with the host cell. If the cell expresses insufficient TMPRSS2 or if the virus–ACE2 complex does not face TMPRSS2, the virus–ACE2 complex is internalized via clathrin-mediated endocytosis. ACE2, angiotensin-2 converting enzyme; TMPRSS2, transmembrane protease serine 2.

**Figure 2 nutrients-14-03303-f002:**
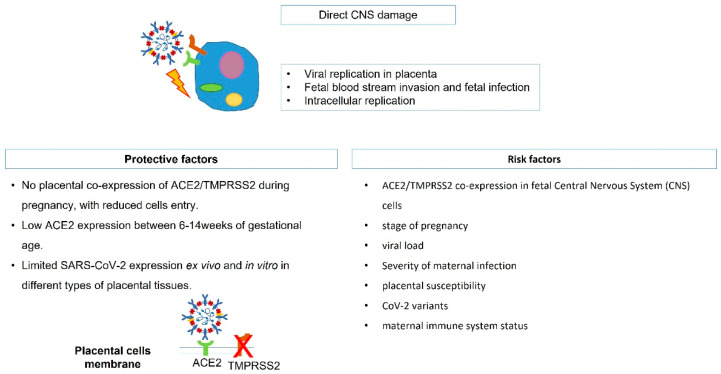
SARS-CoV-2 direct CNS damage mechanism. CNS, central nervous system; ACE2, angiotensin-2 converting enzyme; TMPRSS2, transmembrane protease serine 2.

**Figure 3 nutrients-14-03303-f003:**
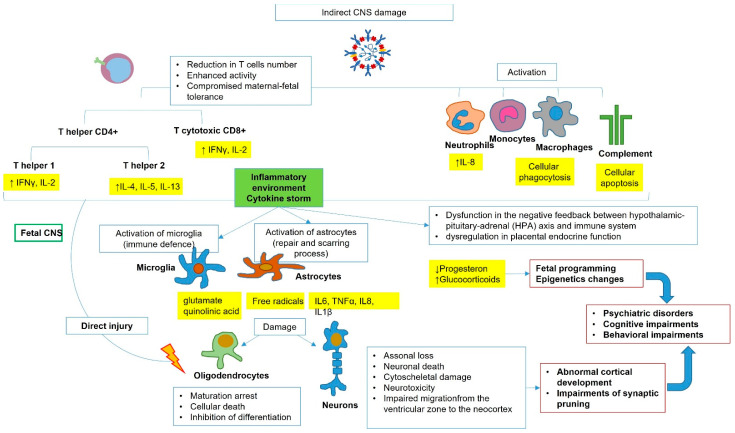
SARS-CoV-2 indirect CNS damage mechanism. CNS, central nervous system. ↓: increase ↑: decrease.

**Table 1 nutrients-14-03303-t001:** Summary of nutrients, sources, suggested daily intake, and potential mechanisms of action.

Nutrients	Sources	RDA in Pregnancy	Action and Properties	References
Amino Acids		71 g/day	- proliferation and activation of T cells, B cells, NK cells, and macrophages; - regulation of intracellular redox status and gene expression - constituent of antibodies, cytokines, and cytotoxic substances	[53]
Arginine	Soy protein, peanuts, walnuts, and fish.	10 g/day	- CoV-2 replication inhibition through nitric oxide synthase enzyme (NOS) and NO production - mediator in neuro-transmission, learning and memory, synaptic plasticity, and neuroprotection	[54,55,56,57,58]
n-3 PUFA	Fish, fish oil, algae oil, walnuts, edible seeds, and flaxseeds	200–300 mg/day	- modulation of inflammation - improve B cell activity and decrease cytokines (TNF-alfa, IL-1beta, IL-6, and IL-8) and inflammatory eicosanoids - improve CD4/CD8 ratio - reduce COX-2 production - decrease viral replication	[59,62,63]
EPA	Fish, fish oil, and algae oil		- decrease prostaglandin E2 (PGE2) production	[61]
DHA	Fish, fish oil, and algae oil		- protection against oxidative insults by inducing inflammatory tolerance	[60]
Fatty Acids (S/MCFA)				
Monolaurin	coconut oil	1–5 mg/day	- dissolving the lipids and phospholipids of the pathogen’s envelope - interference with signal transduction in cell replication - IL-6 and IL-8 decrease - regulation of interferon pathway	[66,67]
Vitamin A (all-trans-retinol)	milk, fish, eggs, carrots, leafy greens, broccoli, cantaloupe, and squash.	770 mcg retinol equivalents (avoid overdoses)	- inhibition of viral replication by upregulating elements of the innate immune response - cofactor in phagocytosis, oxidative processes, and regulation of pro-inflammatory TNF-alpha and IL-2 - regulation of NK cells - development and differentiation of Th1 and Th2 cells	[68,69,70]
Vitamin E (Tocopherol)	olive and sunflower oils (α-tocopherol), soybean and corn oil (γ-tocopherol)	15 mg/day (avoid overdoses)	- antioxidant - reduction in pathogens’ virulence - increase lymphocytes’ proliferation, immunoglobulin levels, and response, NK cells’ activity - modulation of Th1/Th2 balance and IL-2 production - reduction of PGE2 production through decreasing NO levels and inhibiting COX-2	[49,76,77]
Vitamin B1 (thiamine)	whole grains, legumes, meats, and fish	1.4 mg/day	- T cells’ regulation - synthesis of myelin and neurotransmitters - Decrease pro-inflammatory cytokine (Il-1, TNFα, and IL-6) and levels of eicosanoids derived from arachidonic acid	[78,79,80]
Vitamin B6 (pyridoxine)	fruit, vegetables, and grain	1.7 mg/day	- T cells’ and interleukins’ production - lymphocytes’ maturation - synthesis of neurotransmitters	[69]
Vitamin B9 (folic acid)	beef liver, leafy vegetables, peas, beans, avocados, eggs, and milk	-low-risk pregnancies: 0.4 mg/day for 12 weeks -high-risk pregnancies: 4 mg/day in high-risk pregnancy	- immunomodulating properties affecting T cell differentiation - cofactor for the synthesis of purines and thymidylate - role in DNA synthesis and replication	[81,82,83,84,85,86,87]
Vitamin B12 (cobalamin)	meat, clams, liver, fish, poultry, eggs, and dairy products	2.6 µg/day	- DNA synthesis and regulation - role in hemopoiesis - suppress systemic inflammation by modulating IL-6, growth factors - downregulate nuclear transcription factor-kB (NF-kB) involved in immune response to infection - increase CD8 T cells and natural killer T cells against viral infections - antioxidant through the reduced glutathione-sparing effect - analgesic function - role in neuromuscular disorders	[69,88,89,90]
Vitamin C (ascorbic acid)	vegetables and fruits: citrus fruits, strawberries, potatoes, and tomatoes	1.5 g/kg body weight	- protection against oxidative injury - co-factor of eight human enzymes involved in biosynthesis of collagen, carnitine, hormones, and neurotransmitters. - improving neutrophil migration and chemotaxis - modulation of the increase of phagocytic cells - maturation of B and T lymphocytes - support of microbial killing and antibody production - prevention of the excessive activation and accumulation of neutrophils - reduction of epithelial damage	[73,91,92,93,94,95,96,97,98,99]
Iron	red meat, oysters, lentils, beans, poultry, fish, leaf vegetables, watercress, tofu, chickpeas, black-eyed peas, and blackstrap molasses.	27 mg/day	- differentiation and growth of epithelial tissue - production of reactive oxygen species (ROS) by neutrophils - essential for virus replication process	[69,100,101,102]
Zinc	meat, fish, shellfish, fowl, eggs, dairy, wheat (germ and bran), sesame, poppy, alfalfa, celery, mustard, beans, nuts, almonds, whole grains, pumpkin seeds, sunflower seeds, and blackcurrant	11 mg/day	- development, differentiation, and activation of T lymphocytes - secretion of cytokines: IL-1β, IL-6, and TNFα - NK cells’ activity - regulation of NADPH oxidase activity - reduction of ROS generated by pathogens - in high doses, inhibits the activity of RNA-dependent RNA polymerase - inhibition of SARS-Coronavirus replication in cell culture by stopping RNA-synthesizing activity of the multiprotein replication and transcription complex - inhibition of RNA-dependent RNA polymerase elongation and template binding	[49,69,73,105,106,107,108,109,110]
Selenium	meat, nuts (Brazil nuts), cereals, and mushrooms	60 µg/day	- constituent of 25 selenoproteins, with a selenocysteine nucleus, acting as antioxidants (glutathione peroxidase (GPx), involved in control of reactive oxygen species (ROS), deiodinases (Iodothyronine deiodinases), metal chelators (SEPP1), reductases (Thioredoxin reductases, TrxR), modulators of metabolism and insulin sensitivity (SEPS1), and glycoprotein folders (SEP15)) - high selenium levels are associated with pro-inflammatory response	[111,112,113,114,115,116]
Copper	oysters, beef and lamb liver, Brazil nuts, blackstrap molasses, cocoa, black pepper, lobster, nuts and sunflower seeds, green olives, avocados, and wheat bran	1000 μg/day	- role in the functions of T helper cells, B cells, neutrophils, natural killer cells, and macrophages - electron and oxygen transportation - production of ROS by macrophages - inhibition of viral entry and replication - degradation of mRNA and capsid proteins - inactivate some enveloped or non-enveloped, single- or double-stranded DNA or RNA viruses - destroying the viral genomes - disintegration of CoV-2 envelope and dispersal of surface Spikes	[117,118]
Magnesium	Spices, nuts, cereals, cocoa, green leafy vegetables such as spinach, coffee, and tea.	360 mg/day	- control blood glucose level by the inhibition of inflammation, oxidative stress, and smooth muscle contraction - regulation of cytotoxic functions of NK cells and CD8+ T cells - protective against oxidative stress and intracellular glutathione depletion, inflammatory cytokines - neuroprotection of immature fetal brain - high magnesium levels may reduce nitric oxide production and avoid microbicidal activity of macrophages, by enhancing Mg2+-dependent ecto-APT-ase activity	[119,120,121,122,123]
Choline	organ meats, egg yolks, grains, vegetables, fruit, and dairy products	450–550 mg/day	- neurotransmission (as the precursor for the synthesis of acetylcholine) - membrane synthesis (as a basis in the synthesis of phosphatidylcholine) - lipid transport - epigenetic regulation of gene expression (through S-adenosyl methionine synthesis and DNA methylation) - one-carbon metabolism (as a methyl group donor, participating in the re-methylation of homocysteine to methionine) - protective for infant brain development and mitigates the adverse effects of inflammation on offspring’s behavior - decrease fetal brain IL-6 - reduction of inflammation from maternal immune activation - reversing of behavioral effects in offspring	[124,125,126,127,128,129,130,131,132,133,134,135]
Polyphenols	green tea, broccoli, and apples		- natural antioxidant involved in reducing inflammation and immune response	
Curcumin	*Curcuma longa* species plants	3 mg/kg body weight/day	- inhibition of viral replication by binding SARS-CoV-2 main protease (Mpro)/chymotrypsin-like (3CL) - reduction of macrophages population - inhibition of NLR family pyrin domain containing 3 (NLRP3) inflammasome signaling, with reduction in NFkB, TNFα, IL6, IL1β and IL18 expression - reduction of inflammatory pathology - reduction of detrimental effects of chronic glial activation	[136,137,138,139,140]
Flavonoids	red wine, oranges, red fruits and vegetables		- free-radical scavengers and antioxidants - anti-inflammatory and anti-viral effects - inhibition of influenza virus and Toll-like receptor signaling, by blocking NF-KB translocation	[141]
Epigallo-catechin 3 gallate (EGCG)	green tea	120 mg/day	- antibacterial, antiviral, antioxidative, anticancer, and chemo-preventive activities - neuroprotective properties	[63]
Kaempferol	spinach, cabbage, kale and broccoli	5–8 mg/day	- antioxidant and anti-inflammatory properties - high affinity binding with ACE2 receptor with anti-viral activity	[140]
Quercetin	onion, grapes, shallots, tea, Ginko bilboa and tomatoes	500–1000 mg/day	- anti-inflammatory activity by inhibiting lipid peroxidation and lipopolysaccharide-induced IL8 production - immunosuppressive effect on dendritic cells by downregulating T-helper induced IL4	[142,143,144]
Apigenin	parsley, celery, onions and oranges	50 mg/day	- antiviral and anti-inflammatory activity - modulation of dendritic cells responsible for immune balance, decreasing IL6, reducing COX-2 activity - potentially inhibiting SARS-CoV2 viral replication by Mpro binding - reduction of microglia activation in mice	[145]

## Data Availability

Not applicable.

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
