# Peer review of "Fetal Brain Damage during Maternal COVID-19: Emerging Hypothesis, Mechanism, and Possible Mitigation through Maternal-Targeted Nutritional Supplementation"

_nutrients, 2022, doi:10.3390/nu14163303_

Round 1
Reviewer 1 Report
A very interesting and comprehensive narrative review on the effects of SARS-CoV-2 during pregnancy and brain damage. The authors review maternal nutrition on infection and mainly show evidence of the immunologic and inflammatory profile, both maternal and fetal, possibly leading causes of brain damage. Fortunately, and in most of the cases, gestational SARS-CoV-2 infections are low-to-mild and do not induce severe neonatal complications.
I would like to give some comments to consider and improve this review, which is already understandable and easy to read.
1. I would change the title to: "Narrative review of fetal immune-inflammatory effects during gestational COVID19 infection: potential prevention through maternal nutrition."
2. Revise the journal author guidelines, the abstract should not be in italics.
3. Although section 2. is extremely good, I would suggest introducing some kind of figure explaining vertical transmission and its effects on the fetal inflammatory profile.
4. In this same section, are there differences in immune and inflammatory risk depending on fetal sex? I suggest if the authors could review the article: PMID: 32956948
5. perhaps section 3 would be a subsection 2.1.
6. In section 4. I suggest that they introduce a table summarizing the evidence of the text: element / source / % / relationship to cytokine or immune profile / reference. In the text leave only the potential benefit against maternal SARS-coV-2 infection.
7. Revise title section 4.3. and place acronym S/MCFA after "acid".
8. Revise "n3PUFA" and perhaps place section 5. as "conclusions".
Author Response
A very interesting and comprehensive narrative review on the effects of SARS-CoV-2 during pregnancy and brain damage. The authors review maternal nutrition on infection and mainly show evidence of the immunologic and inflammatory profile, both maternal and fetal, possibly leading causes of brain damage. Fortunately, and in most of the cases, gestational SARS-CoV-2 infections are low-to-mild and do not induce severe neonatal complications.
I would like to give some comments to consider and improve this review, which is already understandable and easy to read.
First, we would like to thank Reviewer 1 for suggestion we really appreciate. Herein our answers.
- I would change the title to: "Narrative review of fetal immune-inflammatory effects during gestational COVID19 infection: potential prevention through maternal nutrition."
Thank you. We accept some changes in title, however this review focused on COVID19 effects on fetal brain, thus we prefer to title “Fetal brain damage during maternal COVID19: emerging hypothesis, mechanism and possible mitigation through maternal targeted nutritional supplementation”
- Revise the journal author guidelines, the abstract should not be in italics.
Accepted and changed.
- Although section 2. is extremely good, I would suggest introducing some kind of figure explaining vertical transmission and its effects on the fetal inflammatory profile.
Accepted. We designed a graphical explanation of evidence, named Figure 1, Figure 2 and Figure 3, that we added at the end of manuscript.
- In this same section, are there differences in immune and inflammatory risk depending on fetal sex? I suggest if the authors could review the article: PMID: 32956948
Thanks for suggestion, we add a period regarding this evidence:
“Moreover, a recent study demonstrated that fetal sex may influence the maternal capacity to counteract an inflammatory response: pregnancy with male fetus showed higher levels of IL1β and a greater nitrosative damage in the presence of an oxidative environment, with a possible higher risk of fetal brain damage in male (36)”
- perhaps section 3 would be a subsection 2.1.
Thank you for suggestion, but we prefer to maintain the original numbering.
- In section 4. I suggest that they introduce a table summarizing the evidence of the text: element / source / % / relationship to cytokine or immune profile / reference. In the text leave only the potential benefit against maternal SARS-coV-2 infection.
Thank you for suggestion; we accepted the advice regarding the table, and we produced a table summarizing contents of section 4. We named it Table 1 and we added it to the manuscript together with a period introducing table” Table 1 summarizes evidence for each nutrient considered”. Regarding text, we prefer not to change it, as our aim is also to gather general information about each nutrient.
- Revise title section 4.3. and place acronym S/MCFA after "acid".
Accepted and added.
- Revise "n3PUFA" and perhaps place section 5. as "conclusions".
Accepted. We changed section 5. heading with “Conclusions”.
Please see the attachment.

Reviewer 2 Report
I found this review article rather difficult to evaluate.
There are as yet very few reports of vertical transmission of SARS-COV-2, so whatever effect maternal COVID-19 infection has on the fetal brain will be indirect.
As indicated by the authors in several sections of the article, the possible mechanism of fetal brain injury in maternal COVID-19 infection is most likely non-specific - from inflammatory responses of mother and the fetus, severity of illness in the mother and possibly some impairment of placental function.
I find it such a stretch to attribute brain injury in the babies of COVID-19 infected mothers to the long-list of nutrients reviewed in the article. To suggest that providing such a broad, non-specific nutritional supplementation to COVID-19 infected mothers will somehow prevent brain injury in their babies, to my mind is misleading.
There is very little evidence to support the authors claims and recommendations and I find very little merit in the review article. More tangible evidence needs to accrue before this type of review can be written.
Author Response
Reviewer 2
I found this review article rather difficult to evaluate.
First, we would like to thank Reviewer 2 for his/her time spent to review our article and for his/her observations. Herein our answers.
There are as yet very few reports of vertical transmission of SARS-COV-2, so whatever effect maternal COVID-19 infection has on the fetal brain will be indirect.
As there is still no evidence on fetal brain injury due to SARS-CoV-2 direct cellular damage, we tried to argument these emerging hypothesis starting from data we found in the available literature and findings from other viruses already known. Every clue seems to focus on indirect damage, so we discussed which mechanism could lay under SARS-CoV-2 infection and cells damage, according to the existent knowledge.
As indicated by the authors in several sections of the article, the possible mechanism of fetal brain injury in maternal COVID-19 infection is most likely non-specific - from inflammatory responses of mother and the fetus, severity of illness in the mother and possibly some impairment of placental function. I find it such a stretch to attribute brain injury in the babies of COVID-19 infected mothers to the long-list of nutrients reviewed in the article. To suggest that providing such a broad, non-specific nutritional supplementation to COVID-19 infected mothers will somehow prevent brain injury in their babies, to my mind is misleading.
We absolutely don't attribute fetal brain injury following maternal COVID to nutrients. Conversely, as a potential fetal brain damage following SARS-CoV-2 was hypothesized by many Authors, we start from the possible SARS-CoV-2 indirect mechanisms of cellular damage, including enhanced inflammatory response, cytokine storm, nitrosative damage, immune cells overreaction, asphyxia, that could affect fetal tissue, included fetal central nervous system, and we review a variety of nutrients properties in order find which nutrients may be helpful to mitigate these fetal negative effects.
There is very little evidence to support the authors claims and recommendations and I find very little merit in the review article. More tangible evidence needs to accrue before this type of review can be written.
No claims or recommendations were made by this article; with this review we want to shed lights on an unexplored field and, in particular, on these nutrients that, in our opinion, due to limited evidence and literature, deserve further research, to attest or not a possible positive effect in mitigating COVID negative outcomes on fetal brain.